# Effects of Tree Species Diversity on Fine Root Morphological Characteristics, Productivity and Turnover Rates

Zhibao Wang [1,*], Yongli Cai [2], Jing Liang [3], Qicheng Zhong [3], Hong Jiang [4], Xinghui Lu [1], Xiangbin Gao [1], Shouchao Yu [1] and Xiaojian Dai [1]

1. Agricultural Science and Engineering School, Liaocheng University, Liaocheng 252059, China
2. School of Design, Shanghai Jiao Tong University, Shanghai 200240, China
3. Soil Research Institute, Shanghai Academy of Landscape Architecture Science and Planning, Shanghai 200232, China
4. College of Pharmacy, Guizhou University of Traditional Chinese Medicine, Guiyang 550025, China
* Correspondence: wangzhibao@lcu.edu.cn

**Abstract:** Fine roots ($\varphi \leq 2$ mm) play an important role in the process of material and nutrient cycling in forest ecosystems, but the effect of tree species diversity on the functional characteristics of fine roots is unclear. In this study, $1-7$ subtropical communities with different species richness were selected to study the morphological characteristics, productivity (PRO), and turnover rate (TUR) of fine roots by continuous soil core extraction, ingrowth soil core method, and root analysis system. The effects of tree species diversity on fine root morphological characteristics, PRO, and TUR are also analyzed. The results showed that with the increase in tree species diversity in the community, the effect of fine root morphological characteristics including specific root length (SRL) and specific surface area (SSA) of each community was not significant, but the fine root PRO in the community increased from 71.63 g·m$^{-2}$·a$^{-1}$ (*Ligustrum lucidum* pure forest) to 232.95 g·m$^{-2}$·a$^{-1}$ (*Cinnamomum camphora* mixed forest with seven species richness communities), and the fine root TUR increased from 0.539 times·a$^{-1}$ to 0.747 times·a$^{-1}$. Correlation analysis and redundancy analysis showed that species richness, root functional traits, and soil physicochemical properties were important driving factors affecting root characteristics. The increase in tree species diversity did not change the morphological characteristics of fine roots but increased the PRO and TUR of fine roots.

**Keywords:** fine root morphological characteristics; fine root productivity; fine root turnover rate; species diversity

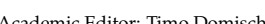



## 1. Introduction

Biodiversity plays an important role in the energy flow and material cycle of the ecosystem [1–6], and the relationship between biodiversity and productivity is a hot issue in current research [7]. As an important biological factor in the ecological environment, biodiversity has a certain impact on the productivity of the community. Most previous studies on the relationship between diversity and productivity are based on grassland ecosystems [8–11] or desert ecosystems [12]. Some scholars believe that productivity increases with the increase in species richness in the community [13–16], but others believe that there are three relationships between productivity and diversity including negative correlation [17–20], S-curve [21], and irrelevant correlation [22,23]. The above research conclusions are based on the above-ground parts of plants. As an important participant, fine roots ($\varphi \leq 2$ mm) also play important roles in carbon allocation and nutrient cycling in ecosystems. Due to the deep distribution of woody plant roots and the difficulty of sampling, there is a lack of research on the impact of plant diversity on ecosystem productivity and turnover rates based on the underground part of plants.

The turnover of fine roots plays an important role in the carbon and nutrient cycle of the ecosystem [24–27]. Plant diversity can have an impact on productivity, and it is bound

to have a certain impact on the turnover rate of fine roots. Although previous scholars have carried out in-depth research on the turnover rate of fine roots, most of them have analyzed the turnover rate of fine roots from the perspective of different environments [28,29] and species [30]. There are few studies on the effect of diversity on the turnover rate of underground fine roots from the perspective of plant diversity. Therefore, the mechanism of the effect of plant diversity on fine root turnover needs to be further explored.

The relationship among plant diversity, fine root productivity, and fine root morphological characteristics is often influenced by abiotic factors such as soil nutrients [9,29,31,32]. The distribution of nutrient resources in soil has a high degree of spatial heterogeneity, and a large number of roots are easily distributed in areas with sufficient nutrients [33–39]. So, scholars believe that the distribution strategy of nutrient resources is an important reason for the increase in root productivity [40]. Soil total phosphorus, soil available nitrogen and available boron [29,30], soil moisture [41,42], temperature [43], pH value [44,45], etc., can affect the fine root productivity of different plant communities. In communities with high species richness, differences in foraging strategies of plant roots and competition among roots lead to root niche differentiation, which allows roots to acquire more resources in a competitive environment, resulting in increased root productivity [13,46–48]. In addition, in order to fully absorb soil resources, the root system will be adjusted according to its own root function characteristics, such as adjusting Specific surface area (SSA), specific root length (SRL), root tissue density (RTD), and root diameter [48–50] in response to nutrient availability. However, our understanding of how root traits predict root productivity and morphological characteristics independently of soil nutrients and forest community properties is still unclear.

So far, more and more studies have explored the effects of biological factors such as tree diversity and species diversity on fine root productivity [13,28,51,52] and fine root morphological characteristics [48–50]. However, fine root research still faces great challenges due to the labor-intensive effort required for fine root sampling and the limitations of identifying fine roots of different tree species in different communities [49,53,54]. In addition, the identification of fine roots of different tree species is often used to visually identify the types of fine roots in mixed samples by their morphological characteristics, such as color, size, odor, and epidermal characteristics [6,51,55]. Due to the above difficulties in the sampling and identification of roots in the community, the current research on plant roots in the community is relatively lacking.

In previous studies on the effect of plant diversity on root productivity and morphological characteristics, most scholars believe that the increase in plant diversity in the community, the intensification of root competition among species, and the niche differentiation of roots of different species are mainly controlled by the hypothesis of niche complementarity [2,56–59]. The niche complementarity hypothesis can be divided into two types: the positive complementarity effect and the negative complementarity effect [19,20,32]. It has been reported that the higher the species richness in the community, the greater the fine root productivity [60,61]. The increase in root productivity is attributable to the strong foraging ability and competition of roots in high species richness communities, leading to the emergence of root niche differentiation, which in turn isolates root space and fully occupies soil space to make full use of soil resources [13,32,62–64]. It will also change the morphological characteristics of the root system [65]. In addition, increased productivity in the community may also be affected by legumes [32]. After the rhizobia of legumes are decomposed, they can increase the N content of the soil and promote root growth. However, some scholars have also found that diversity also has negative complementary effects on the productivity of plant communities, such as antagonism between plants [17,18]. Due to the physical and chemical interference of plants, the productivity of mixed planting is lower than the expected productivity of single planting [56,66]. The above findings were confirmed by comparing the differences in root characteristics in two or three mixed forests with those in a single mixed forest [23,27]. However, with the increase in species richness in the community, how the fine root productivity and morphological characteristics of the

community will change, and what is the reason or mechanism of the change, still needs to be discovered.

Therefore, in this study, seven groups of plant community types with different species richness levels were selected in the coastal area of Shanghai to study the variation of fine root productivity, turnover rate, and morphological characteristics with plant diversity, and to explore the effect of fine roots on soil properties and plant diversity. The following hypotheses were verified through research: (1) when tree species diversity increases, the morphological characteristics of fine roots did not change; (2) with the increase in tree species diversity, the productivity and turnover rate of fine roots accelerated. Based on the above two hypotheses, to explore the effect mechanism of tree species diversity on fine root productivity, turnover rate, and morphological characteristics.

## 2. Materials and Methods

### 2.1. Study Area

The study area is located in Lingang New City, Shanghai (120°53′–121°17′ E, 30°59′–31°16′ N). The region has a subtropical oceanic climate with warm and humid climate, abundant rainfall, and four distinct seasons. The average temperature in the past three years is 15.2 °C–15.8 °C, the average annual precipitation is 900–1050 mm, and 60% of the annual rainfall is mainly concentrated in May–September (Figure 1); the total annual sunshine hours are 2000–2200 h.

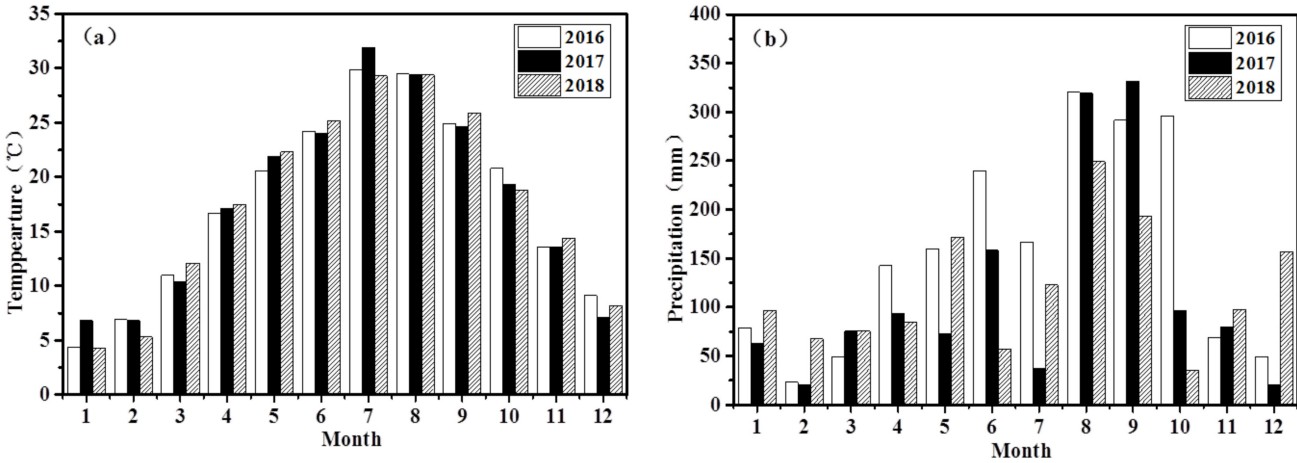

**Figure 1.** Comparison of average temperature and rainfall in Shanghai from 2016 to 2018. (**a**) Average temperature; (**b**) Average rainfall.

This area has high soil salinity ($\geq$0.4%), high soil pH (pH > 8.5), low soil organic matter content (<20 g·kg$^{-1}$); the groundwater level is between 0.5–2.5 m, and the soil moisture content is between 18.07−32.48%; the soil bulk density was between 1.24−1.46 g·cm$^{-3}$, the total nitrogen content was between 0.84 $\pm$ 0.14 g·kg$^{-1}$, and the total phosphorus content was between 1.27 $\pm$ 0.08 g·kg$^{-1}$ [67–69]. The ecological restoration forest planting plan was implemented in February 2011 and completed in April 2011. According to different planting methods, it can be divided into community types composed of different tree species diversity, each type has 5 repetitions. The length and width of this entire experimental site are 900 m and 10 m, respectively. The total area is about 9000 m$^2$.

### 2.2. Setting of Sample Sites

In January 2018, in the experimental area, communities with different species richness composed of 1–7 tree species were selected as the research objects (SR1-SR7), and the average tree height, DBH, crown width, planting density, coverage, and above biomass of the communities were selected (Table 1). In each community type, the area was set as 10 m × 10 m with smooth surface each plot, and the plots were surrounded by 1 m × 1 m root-isolating boards. The thickness of the understory litter layer was 1 cm, and no herbs

grew; 5 parallel plots were set for each community type, and the interval between each plot is more than 50 m, in order to avoid edge effects of other plots.

**Table 1.** Characteristics of plant communities within different tree species richness.

| Species Richness | Tree Species | Height (m) | DBH (cm) | Crown Width (m) | Ratio (%) | Coverage (%) | Stand Density (Trees·hm⁻¹) |
|---|---|---|---|---|---|---|---|
| SR1 | *Ligustrum lucidum* | 5.43 ± 0.12 | 5.02 ± 0.47 | 2.84 ± 1.53 | 100% | 96 | 4000 |
| SR2 | *Ligustrum lucidum* | 5.58 ± 1.08 | 5.85 ± 0.31 | 3.13 ± 1.75 | 50% | 96 | 4000 |
| | *Melia azedarach* | | | | 50% | | |
| SR3 | *Ligustrum lucidum* | 5.16 ± 1.61 | 5.74 ± 0.59 | 3.14 ± 1.91 | 33.33% | 96 | 4000 |
| | *Sapium sebiferum* | | | | 33.33% | | |
| | *Quercus virginiana* | | | | 33.33% | | |
| SR4 | *Ligustrum lucidum* | 5.43 ± 3.12 (1.35 ± 0.10) | 6.02 ± 4.17 (0.52 ± 0.10) | 2.84 ± 1.73 (0.45 ± 0.10) | 33.00% | 96 | 4000 |
| | *Populus 'Zhonghua Hongye'* | | | | 33.00% | | |
| | *Cinnamomum camphora* | | | | 33.00% | | |
| | *Clerodendrum cyrtophyllum* | | | | 1.00% | | |
| SR5 | *Ligustrum lucidum* | 5.11 ± 3.34 (1.57 ± 0.30) | 5.31 ± 5.23 (1.12 ± 0.20) | 3.16 ± 1.87 (0.5 ± 0.2) | 24.75% | 96 | 4000 |
| | *Broussonetia papyrifera* | | | | 24.75% | | |
| | *Populus 'Zhonghua Hongye'* | | | | 24.75% | | |
| | *Cinnamomum camphora* | | | | 24.75% | | |
| | *Clerodendrum cyrtophyllum* | | | | 1.00% | | |
| SR6 | *Ligustrum lucidum* | 4.75 ± 3.45 (1.53 ± 0.13) | 5.42 ± 5.25 (1.37 ± 0.20) | 3.00 ± 1.87 (1.14 ± 0.10) | 19.80% | 96 | 4000 |
| | *Salix matsudana* | | | | 19.60% | | |
| | *Photinia fraseri* | | | | 19.80% | | |
| | *Robinia pseudoacacia* | | | | 19.80% | | |
| | *Viburnum odoratissimum* | | | | 19.80% | | |
| | *Eurya emarginata* | | | | 1.00% | | |
| SR7 | *Ligustrum lucidum* | 4.35 ± 2.80 (1.62 ± 0.18) | 5.02 ± 3.32 (1.07 ± 0.20) | 2.15 ± 0.90 (0.95 ± 0.10) | 16.50% | 96 | 4000 |
| | *Euonymusbungeanus* | | | | 16.50% | | |
| | *Cinnamomum camphora* | | | | 16.50% | | |
| | *Salix matsudana* | | | | 16.50% | | |
| | *Sapium sebiferum* | | | | 16.50% | | |
| | *Robinia pseudoacacia* | | | | 16.50% | | |
| | *Eurya emarginata* | | | | 1.00% | | |

### 2.3. Sequential Coring

In January, July, and November 2018, fine roots were sampled with a steel bucket-type soil auger ($\varphi$ = 5 cm, H = 30 cm) with a T-handle in each plot. The "S" shaped 9-point sampling was used with a sampling depth of 50 cm [70]. Root and soil samples were collected in five different layers of 0–10 cm, 10–20 cm, 20–30 cm, 30–40 cm, and 40–50 cm, respectively. After the roots in the same soil layer of the same community type were evenly mixed, three parallel samples were formed. After that, the samples were put into Ziplock bags and brought back to the laboratory, and stored in the refrigerator at 4 °C for 1 month. When sampling the root system, the time without obvious precipitation was selected to avoid the influence of precipitation factors on the experimental results.

### 2.4. Root Ingrowth Cores

Ingrown soil core method is the root biomass that grows in a certain volume of unrooted soil within a certain period of time [71–73]. The growth of fine roots was used to calculate the productivity of fine roots. Due to the lag period of root growth [25,74], the experiment of in-growth soil core method was implemented in November 2017. In each community type, a root drill ($\varphi = 5$ cm, H = 30 cm) was used to drill the soil core with a sampling depth of 50 cm. Nine growing soil cores were set for each plot of each type (five parallel plots for each community type), and 315 growing soil cores were set for each of the seven community types. After that, all the roots in the soil core were sieved with a sieve ($\varphi = 2$ mm), and the soil without roots was put into a nylon mesh bag (L = 50 cm, $\varphi = 5$ cm) with an aperture of 0.15 cm. Then the nylon bag was placed in the soil core cavity [75], and the position of the soil core was marked with PVC pipe ($\varphi = 5$ cm, H = 1 cm). All set ingrown bags were removed in two separate batches in July and December 2018, respectively. When taking the soil core, the full excavation method is used to collect the roots in the soil core according to five layers of 0–10 cm, 10–20 cm, 20–30 cm, 30–40 cm, and 40–50 cm. The collected roots are quickly put in a plastic bag and bring it back to the laboratory for refrigerated storage (4 °C). All root samples are processed within 1 month.

### 2.5. Fine Root Isolation and Measurement

Rinse the entire soil on the root surface with clean water in the laboratory. Use absorbent paper to dry up the water on the surface of the root system, and then use tweezers and vernier calipers to screen out fine roots with $\varphi \leq 2$ mm [76]. After the grass roots are removed, the living and dead fine roots of different tree species are distinguished according to the shape, color, smell, and elasticity of the fine roots [77,78]. The treated fine root samples were placed in a transparent scanner tray, and the root system scanning analyzer Win-RHIZO 2005C (Regent Instruments Inc., Quebec, QC, Canada) was used to scan the fine root samples to obtain the average diameter (D), surface area (S), volume (V) of and length (L) and the fine roots other data. Based on the above data, calculate root length density (RLD), specific root length (SRL), specific surface area (SSA), root tissue density (RTD), root surface area density (RSAD), and other indicators [79]. Finally, the fine root samples were dried in an oven at 80 °C for 24 h to a constant weight, and the fine root biomass (FRB) was calculated. Calculated as follows:

$$\text{FRB (g·m}^{-3}) = g \times 10^4 / [\pi(d/2)^2] \tag{1}$$

$$\text{RLD (m·m}^{-3}) = l \times 10^6 / [\pi(d/2)^2 \times h] \tag{2}$$

$$\text{SRL (m·g}^{-1}) = l/g \tag{3}$$

$$\text{SSA (cm}^2\text{·g}^{-1}) = s/g \tag{4}$$

$$\text{RTD (g·m}^{-3}) = g/v \tag{5}$$

$$\text{RSAD (m}^2\text{·m}^{-3}) = s \times 10^6 / [\pi(d/2)^2 \times h] \tag{6}$$

In the formula: g, fine root dry weight (g); l, total length of fine root (cm); d, diameter of soil auger (cm); h, height of soil auger (cm); s, fine root surface area (cm$^2$); v, volume of fine root (cm$^3$);

### 2.6. Fine Root Productivity (PRO) and Turnover Rate (TUR)

The biomass of fine roots growing in the unrooted soil core in a certain period of time is regarded as the net production of fine roots in this period of time, that is, the production of fine roots [26]. In this study, the productivity of the root system of the community was

calculated by measuring the biomass of fine roots growing in the soil core over 12 months, and the calculation was as follows:

$$PRO = FR_L + FR_D \tag{7}$$

Among them, PRO is the annual net productivity of fine roots ($g \cdot m^{-2} \cdot a^{-1}$), and $FR_L$ and $FR_D$ are the living and dead fine root biomass in the unrooted soil column within 12 months.

Turnover rate of fine roots [28,80]:

$$TUR = PRO/Y \tag{8}$$

TUR is the turnover rate of fine roots ($times \cdot a^{-1}$); PRO is the annual net productivity of fine roots, and Y is the average biomass of living fine roots (the mean value of living fine root biomass measured by the continuous soil drilling method in this study) [81].

### 2.7. Physical and Chemical Properties of Soil

At the same time as fine roots were collected from each plot, the conductivity, temperature, and pH of each soil layer (0–10 cm, 10–20 cm, 20–30 cm, 30–40 cm, 40 cm) were measured with a calibrated portable conductivity meter EC Tester 11+ (Spectrum Technologies Inc., Aurora, IL, USA) and pH meter (Spectrum Technologies Inc., Aurora, IL, USA). In each soil layer, the upper, middle, and lower parts of the soil core were measured, respectively, that is, the conductivity, temperature, and pH were measured three times, and then the average value was taken. At the same time, the soil of each soil layer was collected and put into 3 aluminum boxes, respectively. After being sealed with Ziplock bags, they were brought back to the laboratory. Finally, the soil moisture (SM) content was measured by the drying method. According to the method of Bao (2005) [82], indicators such as soil organic matter (SOM), total nitrogen (TN), and total phosphorus (TP) were measured in the laboratory for each experimental plot.

### 2.8. Data Processing

SPSS (Statistical Product and Service Solutions) 16.0 software was used for statistical analysis, and all data were tested for homogeneity of variance before statistical analysis. The logarithmic transformation was performed if the variance was unequal. One-way analysis of variance (one-way ANOVA) and least significant difference (LSD) were used to analyze the differences in fine root productivity, turnover rate and fine root morphology in different communities. In addition, multivariate variance analysis was used to analyze the effects of different communities, soil depths and their interaction on fine root productivity, turnover rate and fine root morphological characteristics. Statistical analysis significance level $p \leq 0.05$. Pearson correlation analysis was used to determine the relationship between fine root index and soil, while RDA two-dimensional ordination map is analyzed by R software. The rest of the analysis graphs were drawn using Origin Pro 9.0 software.

## 3. Results

### 3.1. Morphological Characteristics of Fine Roots in Communities with Different Species Richness

Species richness, soil depth, and the interaction between soil depth and species richness had no significant effects on RTD, SRL, and SSA ($p > 0.05$, Table 2), indicating that the fine roots of tree species in the community did not respond to interspecific competition through morphological plasticity. While species richness, soil depth, and the interaction of species richness and soil depth had significant effects on RLD and RSAD, respectively ($p < 0.05$).

**Table 2.** Results of ANOVA of the effects of tree species richness and soil layers on fine root morphological characteristics.

| Parameter | Source of Variation | | |
|---|---|---|---|
| | Tree Species Richness | Soil Depth | Tree Species Richness × Soil Depth |
| RTD (g·cm$^{-3}$) | 1.68 | 4.02 | 1.26 |
| RLD (m·m$^{-3}$) | 3.84 * | 20.79 ** | 2.14 * |
| SRL (m·g$^{-1}$) | 0.46 | 0.92 | 1.05 |
| SSA (cm$^2$·g$^{-1}$) | 0.50 | 0.62 | 0.96 |
| RSAD(cm$^2$·m$^{-3}$) | 5.39 ** | 21.69 ** | 2.34 ** |

Note: ** At 0.01 level (double side) significant difference, * at 0.05 level (double side) significant difference.

The RLD and RSAD of each community increased gradually with the increase in species richness (Figure 2b,e). Among them, the RLD and RSAD of SR7 are the largest, which are 35,302.42 m·m$^{-3}$ and 36.81 m$^2$·m$^{-3}$, respectively, while the RLD and RSAD of SR1 are the smallest, presenting 13,038.58 m·m$^{-3}$ and 12.98 m$^2$·m$^{-3}$, respectively. Meanwhile, in different soil layers, the RLD and RSAD of each community showed a decreasing trend with the increase in soil depth (Figure 2b,e). Among them, the difference was significant in the 0–10 cm layer ($p < 0.05$), and the difference was small in other soil layers ($p > 0.05$). However, in the 0–50 cm soil layer, the RLD and RSAD of different species richness communities were significantly different ($p < 0.05$) (Table 2).

The root tissue density (RTD), specific root length (SRL), and specific surface area (SSA) of fine roots in each community varied with soil depth (Figure 2a,c,d), while the difference in total was not significant ($p > 0.05$, Table 2). Among them, the SRL and SSA of SR1 increased gradually with the increase in the soil layer, while the RTD showed a gradually decreasing trend. The SRL and SSA of SR2 decreased gradually with the increase in the soil layer. For RTD, it has a gradually increasing trend with the increase in the soil layer depth. The SRL and SSA of SR3 reached the maximum at the 10−20 cm layer and reached the minimum at the 30−40 cm layer. The RTD of SR3 gradually increased with the increase in soil depth. The SRL and SSA of SR4 increased gradually with the increase in soil depth. The RTD of SR4 first increased and then decreased with the increase in soil depth, reaching the maximum value in the 30−40 cm layer. The SRL and SSA of SR5 gradually decreased with the increase in soil depth, and RTD showed irregular fluctuations in different soil layers. Moreover, the RTD, SRL, and SSA of SR6 and SR7 presented irregular fluctuations in different soil layers (Figure 2a,c,d).

*3.2. Fine Root Production and Turnover in Communities of Different Species Richness*

The effect of species richness on fine root productivity and turnover rate was significant ($p < 0.05$, Table 3). While the effect of soil depth on fine root productivity and turnover rate was not significant ($p > 0.05$). Meanwhile, the interaction effect on fine root productivity and turnover rate was also small ($p > 0.05$). It can be seen that the fine root productivity and turnover rate were mainly affected by species richness in the community.

Among the seven groups of communities with different species richness, SR7 was the largest at 232.95 g·m$^{-2}$·a$^{-1}$; SR1 was the smallest at 71.63 g·m$^{-2}$·a$^{-1}$; the fine root productivity of SR2-SR6 centered at 82.34 g·m$^{-2}$·a$^{-1}$, 94.99 g·m$^{-2}$·a$^{-1}$, 101.36 g·m$^{-2}$·a$^{-1}$, 133.75 g·m$^{-2}$·a$^{-1}$, and 148.04 g·m$^{-2}$·a$^{-1}$, respectively. In conclusion, with the increase in species richness in the constituent community, the fine root productivity gradually increased (Figure 3a).

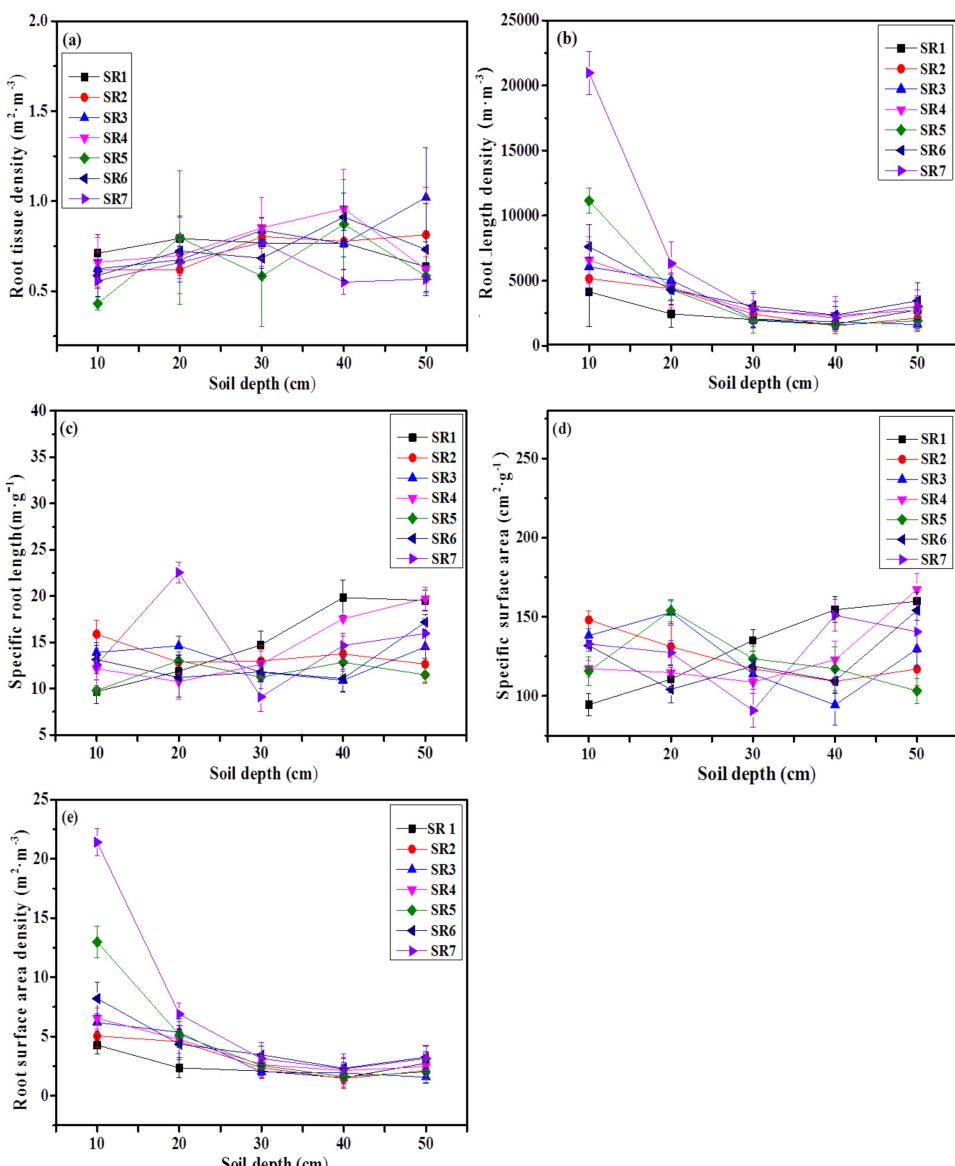

**Figure 2.** Morphological characteristics of fine roots in plant communities with different tree species richness. (**a**) Root tissue density; (**b**) Root length density; (**c**) Specific root length; (**d**) Specific surface area; (**e**) Root surface-area density. SR1, *Ligustrum lucidum* monoculture forest; SR2, *Melia azedarach* mixture forest; SR3, *Sapium sebiferum* mixture forest; SR4, *Populus deltoids* mixture forest; SR5, *Broussonetia papyrifera* mixture forest; SR6, *Salix matsudana* mixture forest; SR7, *Cinnamomum camphora* mixture forest. The same below.

**Table 3.** Results of ANOVA of the effects of tree species richness and soil layers on the annual productivity and turnover rate of fine roots.

| Parameter | Source of Variation | | | | | |
| | Species Richness | | Soil Depth | | Species Richness × Soil Depth | |
| | *F* | *p* | *F* | *p* | *F* | *p* |
|---|---|---|---|---|---|---|
| Productivity (g·m⁻²·a⁻¹) | 8.55 | <0.01 ** | 1.90 | Ns | 0.85 | Ns |
| Turnover rates (times·a⁻¹) | 5.36 | <0.05 * | 0.49 | Ns | 0.46 | Ns |

Note: ** At 0.01 level (double side) significant difference, * at 0.05 level (double side) significant difference.

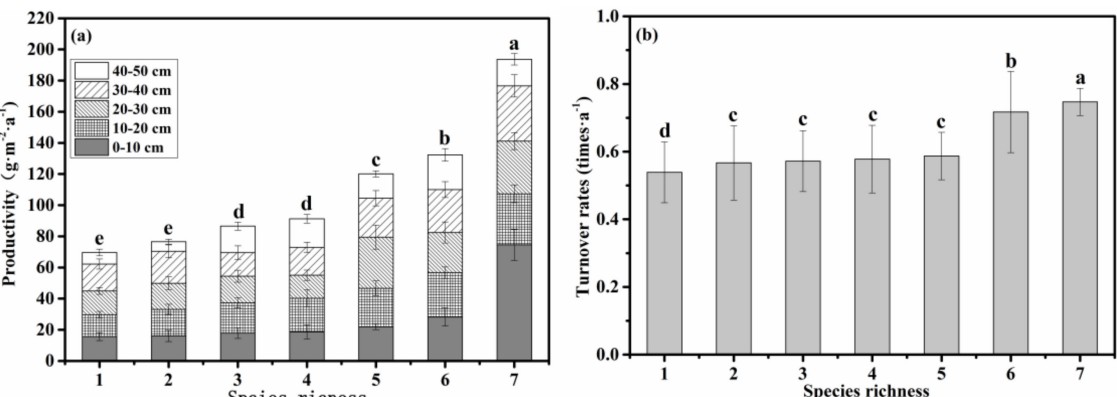

**Figure 3.** Fine root annual productivity and turnover rates in plant communities with different tree species richness. (**a**) Productivity; (**b**) Turnover rates; Note: different letters indicate significant differences, and the same letters indicate no significant difference ($p < 0.05$).

The fine root turnover rates of the seven groups of communities with different species richness were: 0.539 times·a$^{-1}$ for SR1, 0.567 times·a$^{-1}$ for SR2, 0.572 times·a$^{-1}$ for SR3, 0.578 times·a$^{-1}$ for SR4, 0.587 times·a$^{-1}$ for SR5, 0.717 times·a$^{-1}$ for SR6, and 0.747 times·a$^{-1}$ for SR7. Overall, the fine root turnover rate increased gradually with the increase in species richness in the community (Figure 3b).

*3.3. Relationship between Fine Root Characteristics and Environmental Factors in Communities with Different Species Richness*

Through redundancy analysis and correlation analysis of fine root characteristics and environmental factors of different species richness communities, the RDA two-dimensional ordination map (Figure 4) and correlation analysis (Table 4) were obtained. The research showed that: RTD, RLD, SRL, and SSA had little correlation with species richness ($p > 0.05$), while RSAD had a significant positive correlation with species richness ($p < 0.05$). RTD, RLD, SRL, SSA, and RSAD were not correlated with pH, SM, TN, and TP ($p > 0.05$). RTD was significantly negatively correlated with T and SOM ($p < 0.01$), while RLD and RSAD were extremely significantly negatively correlated with EC ($p < 0.01$), and significantly positively correlated with T and SOM ($p < 0.01$).

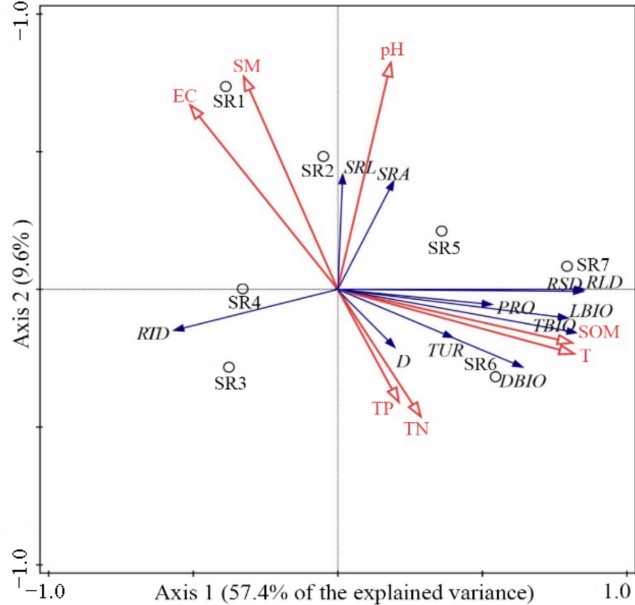

**Figure 4.** RDA dimensional sequencing diagram.

**Table 4.** The correlation between fine root characteristics and soil physical and chemical properties in different communities.

| Indicator | SR | pH | EC | SM | T | TN | TP | SOM | PRO | TUR | D | RTD | RLD | SRL | SSA | RSD |
|---|---|---|---|---|---|---|---|---|---|---|---|---|---|---|---|---|
| PRO | 0.735 ** | 0.105 | −0.401 * | −0.405 ** | 0.450 ** | −0.103 | −0.089 | −0.284 | 1 | | | | | | | |
| TUR | 0.342 * | 0.167 | −0.406 * | 0.116 | 0.395 ** | 0.424 * | 0.336 * | 0.464 ** | 0.248 | 1 | | | | | | |
| D | 0.258 | 0.088 | −0.434 ** | −0.387 * | 0.241 | 0.128 | 0.136 | 0.308 | 0.438 ** | 0.211 | 1 | | | | | |
| RTD | −0.269 | −0.050 | 0.165 | 0.020 | −0.480 ** | −0.181 | −0.227 | −0.435 ** | −0.309 | 0.085 | 0.085 | 1 | | | | |
| RLD | 0.328 | −0.260 | −0.436 ** | −0.278 | 0.697 ** | 0.245 | 0.180 | 0.693 ** | 0.678 ** | 0.399 * | −0.399 ** | −0.479 ** | 1 | | | |
| SRL | −0.078 | 0.306 | 0.155 | 0.250 | −0.099 | −0.351 | −0.291 | −0.149 | −0.050 | 0.091 | 0.091 | −0.008 | −0.095 | 1 | | |
| SSA | −0.043 | 0.203 | 0.155 | 0.194 | 0.063 | −0.205 | −0.138 | −0.036 | 0.001 | 0.084 | 0.084 | −0.261 | 0.089 | 0.720 ** | 1 | |
| RSD | 0.346 * | −0.255 | −0.432 * | −0.274 | 0.691 ** | 0.251 | 0.212 | 0.693 ** | 0.684 ** | 0.402 * | −0.402 ** | −0.522 ** | 0.996 ** | −0.121 | 0.077 | 1 |

Note: SR: species richness; EC: electrical conductivity; SM: soil moisture; T: temperature; SOM: soil organic matter; TN: total N; TP: total P; PRO: productive; TUR: turnover; D: fine root diameter; RTD: root tissue density; RLD: root length density; SRL: specific root length; SSA: specific surface area; RSD: root surface area density. * At 0.05 level (double side) significant correlation, ** at 0.01 level (double side) significant correlation.

There was a significant positive correlation between fine root productivity and species richness ($p < 0.05$), a significant positive correlation with RLD and RSAD ($p < 0.05$), no correlation with RTD, SRL and SSA ($p > 0.01$), a significant negative correlation with EC and SM ($p < 0.05$), a significant positive correlation with T ($p < 0.05$), and no correlation with TN, TP and SOM ($p < 0.05$).

For fine root turnover, it had a significant positive correlation with species richness ($p < 0.05$), a significant positive correlation with RLD and RSAD ($p < 0.05$), no correlation with RLD, SRL and SSA ($p < 0.05$), a significant negative correlation with soil conductivity EC ($p < 0.05$), a significantly positively correlated with T, TN, TP and SOM ($p < 0.05$), and not correlated with soil pH and SM ($p < 0.05$).

## 4. Discussions

### 4.1. Effects of Tree Species Diversity on Morphological Characteristics of Fine Roots

Fine root morphology has a high degree of plasticity. When limited by soil nutrients or interspecific competition, the ability to acquire nutrients or competitiveness can be improved by increasing RTD, SRL, SSA, or branching mode [83–87]. In this study, the RTD, SRL, and SSA of fine roots of seven different species richness communities were not significantly different ($p > 0.05$, Table 2). They also did not increase with the enlargement of tree species diversity. It can be considered that they increased with the rise of species richness in the community. When the morphological characteristics of fine roots are changed, the plant does not respond to root competition, which is consistent with Meinen's findings (2009) and does not support the first hypothesis proposed above [23].

The changes in RLD and RSAD in the community were mainly affected by the biomass of fine roots per unit area [88]. There was no significant difference in the morphological characteristics of fine roots among the communities, which may be due to: (1) differences in fine root biomass among different communities just offset the differences in fine root morphological characteristics of stands [81]; (2) there were no significant differences in soil water and nutrient resources among the plots, so soil resources did not have a certain impact on root foraging behavior; (3) the changes in species composition and root characteristics in the community may mask the effect of the species' genetic characteristics on the morphological characteristics of fine roots, resulting in the insignificant effects of species richness on RTD, SRL, and SSA of fine roots ($p > 0.05$). This indicated that tree species diversity did not affect the morphological characteristics of fine roots.

### 4.2. Effects of Tree Species Diversity on Fine Root Productivity and Turnover

Previous studies have confirmed that the root system of a mixed forest is more competitive than that of a pure forest. Root competition can significantly affect root productivity, and the amount of soil resources available to plants in a competitive environment directly affects the size of fine root productivity [89].

This study showed that the fine root productivity of the pure privet forest was the smallest ($71.63 \ \mathrm{g \cdot m^{-2} \cdot a^{-1}}$), the fine root productivity of the community composed of 2–5 tree species was the middle, and the fine root productivity of the camphor mixed forest composed of seven species is the highest, reaching $232.95 \ \mathrm{g \cdot m^{-2} \cdot a^{-1}}$. The differences in fine root productivity among communities with different species richness were significant and gradually increased with the increase in tree species diversity (Figure 3a), reflecting the positive effect of diversity on fine root productivity [28]. The results support the second hypothesis of this paper. The conclusion that species richness is positively correlated with fine root productivity is also validated in many economic forests or forest types of different forest ages [13,15,60]. For example, Lei et al. (2012) believed that species richness had a positive impact on community productivity, and the main reason was that in species-rich forests, fine roots had strong anti-disturbance and regeneration abilities [15].

Differences in root characteristics (deep-rooted, shallow-rooted) and functional traits (resource-conserving, resource-acquisitive) of different tree species in the community may also positively affect fine root productivity [27]. The root system of the nitrogen-fixing

plant *Robinia pseudoacacia* in the community is deep, and it is a deep-rooted tree species with small root diameter, the largest SRL and SSA, and the strongest ability to acquire soil resources, so is a resource-acquiring tree species. Virginia oak has the largest diameter, the smallest SRL and SSA, belonging to resource-conservative tree species. At the same time, tree species with different life forms (evergreen trees, deciduous trees or shrubs) were added to the communities with different species richness, and the functional trait diversity of the roots of tree species in the whole community increased, leading to the enhancement of the competition between roots. The distribution characteristics of fine roots of different tree species in the community and the competition mechanism between species can complement the root ecosystem, improve the utilization efficiency of soil resources, and increase the fine root productivity of the community [30,59]. Other scholars have also reached consistent conclusions on the impact of plant diversity on productivity. For example, Loreau (2004) believed that the roots of different species in the community have a certain complementarity and can occupy non-overlapping ecological niches in the soil [90]. The expansion of niche space can make full use of water and nutrient resources in different soil layers, which is an important reason for the increase in fine root productivity.

Fine root turnover is an important process involved in carbon and nutrient cycling in forest ecosystems [91–94]. The results show that the turnover rate of fine roots is easily affected by forest types [24,93]. The turnover rate is usually in the range of 0.29–1.20 times·a$^{-1}$, and most of them are between 0.5–1.20 times·a$^{-1}$. In this study, the order of fine root turnover rate in 7 different tree species diversity communities was as follows: SR7 (0.747) > SR6 (0.717) > SR5 (0.587) > SR4 (0.578) > SR3 (0.572) > SR2 (0.567) > SR1 (0.539) (unit: times·a$^{-1}$). The findings support the second hypothesis: fine root turnover increases as tree species diversity rises. The fine root turnover rates of the above communities were all in the range of 0.539–0.747 times·a$^{-1}$, which was within the range of Shan et al. (1993) for the fine root turnover rates of different tree species (0.47–1.05 times·a$^{-1}$) [95]. In this study, the fine root turnover rate of the community composed of 2–5 tree species was not significantly different ($p > 0.05$, Figure 3b), while the fine root turnover rate of SR6, SR7, and SR1-SR4 was significantly different ($p < 0.05$), which may be related to the role of *Robinia pseudoacacia* in the community. As a nitrogen-fixing plant, *Robinia pseudoacacia* can provide a lot of nutrients for plants to grow and plants have increased fine root circumferences in nutrient-rich soils. In addition, the increase in tree species diversity and the intensification of root competition among plants lead to an increase in fine root mortality and an increase in fine root turnover [29].

### 4.3. Effects of Biotic and Abiotic Factors on Fine Root Morphological Characteristics, Productivity and Turnover

Fine root productivity, turnover and morphological characteristics are affected by biological factors [48,49,51]. Among the fine root morphological characteristics, except RTD, D, RTD, RLD, SRL, and SSA were not correlated with species richness, that is, species richness did not have a significant impact on the above fine root morphological characteristics, indicating that as the diversity of tree species in the community increased, the plant does not respond to root competition although the morphological characteristics of fine roots are changed. There was a very significant positive correlation between fine root productivity and species richness [60,61]. In communities with high species richness, there were differences in root characteristics (deep root and shallow root) of different tree species. At the same time, the roots of different tree species occupy different ecological niches in the soil, so the complementarity of root space can be achieved [2,32,57,59,61,64]. In addition, different plant root combinations can fully absorb water and nutrients in the soil, resulting in an increase in the fine root productivity of tree species with an increase in tree species diversity [60,61]. Moreover, in tree species-rich communities, the diversity of tree canopy structures can promote the full utilization of sunlight by different tree species, and trees can distribute more photosynthetic products to the ground, increasing the productivity of underground fine roots. There is a significant positive correlation between the turnover

rate of fine roots and species richness, indicating that the increase in tree species diversity can accelerate the turnover rate of fine roots. The main reason is that the competition of roots of different species in the community leads to the increase in dead fine roots, which in turn accelerates the turnover of fine roots [29].

Fine root morphological characteristics, productivity, and turnover are also affected by abiotic factors [42,43,48,50]. Among the fine root morphological indicators, D was significantly negatively correlated with EC and SM; RTD was extremely significantly negatively correlated with T and SPM; RLD and RSAD were both extremely significantly positively correlated with T and SOM, and extremely significantly negatively correlated with EC. This shows that in the area with a suitable environment or sufficient nutrients, the root system is resource-acquisitive, and the root system has a strong foraging ability; in the nutrient-poor area, the plant root system is resource-conservative, so as to adapt to the soil environment [27,48,49]. The fine root productivity was significantly positively correlated with T, extremely significantly negatively correlated with SM, negatively correlated with EC, and not correlated with pH, TN, TP, and SOM, indicating that T, SM, and EC are the key environments affecting fine root productivity factor [42,43]. The fine root turnover rate was significantly positively correlated with T, TN, TP, and SOM, significantly negatively correlated with EC, and not correlated with pH, illustrating that soil nutrients and EC significantly affected the fine root turnover rate. In summary, soil physicochemical properties are important factors affecting the characteristics of fine roots. This conclusion is consistent with that of Xu et al. (2019) [29] and Zeng et al. (2019) [30].

In addition, the morphological characteristics, productivity, and turnover rate of fine roots were also affected by their own functional traits. In this study, both D and RTD had a very significant negative correlation with RLD and RSAD; RLD had a very significant positive correlation with RSAD, and SRL had a very significant positive correlation with SSA. The fine root productivity and turnover rate were significantly positively correlated with D, RLD, and RSAD. This indicated that fine root functional traits were also important factors affecting fine root characteristics [6,96,97].

## 5. Conclusions

The results of this study showed that with the increase in tree species diversity in the community, the fine root morphology (specific root length and specific surface area) of each community did not change significantly, that is, fine roots did not respond to interspecific competition through morphological plasticity. With the increase in tree species diversity in the community, the competition between fine roots intensifies, and fine roots make more full use of nutrient resources in different soil layers, resulting in the phenomenon of niche differentiation of fine roots. Due to the role of fine root competition, resource utilization strategies, and niche differentiation, the fine root productivity in the community increased and the turnover rate accelerated. This verifies the positive effects of complementarity on fine root productivity and turnover in communities with different tree species diversity. Species richness, root functional traits, and soil physicochemical properties are important driving factors affecting fine root characteristics.

**Author Contributions:** Y.C., Z.W. and H.J. conceived the idea and Z.W., H.J. designed the research; data collection and analyses were performed by Z.W., X.L. and X.G.; statistical analyses were performed by Z.W., J.L., S.Y. and X.D. with input from S.Y., Q.Z. The manuscript was written by Z.W., Y.C., and H.J., with input from X.G., and Q.Z. All authors have read and agreed to the published version of the manuscript.

**Funding:** This study was funded by the National Natural Science Foundation of China (31901210) and Liaocheng University Innovation and Entrepreneurship training program (CXCY2022016, CXCY2022092, CXCY2022372), and the Doctoral Research Project of Liaocheng University (318052123). Thanks to Changlu Wu, School of Life Sciences, Fudan University, for proofreading this paper.

**Institutional Review Board Statement:** Not applicable.

**Informed Consent Statement:** Not applicable.

**Data Availability Statement:** The entire datasets used and/or analyzed in the study are available from the corresponding author.

**Conflicts of Interest:** The authors declare no conflict of interest.

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
