# Peer review of "Effects of Tree Species Diversity on Fine Root Morphological Characteristics, Productivity and Turnover Rates"

_forests, doi:10.3390/f13101740_

Round 1
Reviewer 1 Report
The manuscript entitled “Effects of Tree Species Diversity on Fine Root Morphological Characteristics, Productivity and Turnover rates” is assessing effect of tree species diversity on fine root productivity, turnover rate and morphological characteristics. The manuscript is interesting for the readers, however, it is not a complete paper, so it must improve and needs major revision. The most important part that must improve is discussion and novelty of the paper. Several major and minor comments that help to improve the paper as follow:
Line 40à what is he role of roots in ecosystems?
Line 41-43à Several studies conducted about the roots and their different roles in ecosystem such as slope stability, hydrology, Biomass, etc. So, it is right that we still have a lack of knowledge, but it is not a good novelty for your study.
Line 44à reference?
Line 54à Only soil nutrients? Influenced by biotic and abiotic.
Line 55-56à distribution of roots depends on different factors such as species age, vegetation species (trees, shrubs, herbaceous), topography, growth conditions (including soil temperature, soil depth, nutrient), and moisture content etc. Here you can add these parameters and cite the most recent studies such as:
https://doi.org/10.3390/f11030345
https://doi.org/10.1016/j.ecoleng.2017.04.034
https://doi.org/10.1007/s40333-018-0021-2
https://doi.org/10.3390/f10040341
https://doi.org/10.1016/j.foreco.2020.118873
https://doi.org/10.33494/nzjfs502020x68x
https://doi.org/10.1016/j.catena.2022.106410
Line 94-95à Reference?
Line 99-102à Actually, root morphology also depends on the topography and genetic or tree species. It is not true that species richness is the only parameter that affect root morphology.
The novelty of the study is not clear. Clarify it.
The main and secondary aims of the study is not clear.
Line 122-122à Did you do a soil experiment? If yes, explain the methods of sampling and number of samples. If not, how do you have the values of different soil properties?
Line 131à What are the species?
Line 134à What were the species that selected for the study?
Line 136à “As shown in Table 1.” Do not use short sentences.
Why write the species scientific name as a note again?
Line 139à How is it possible that herbs do nor grow in the soil?
What are the “ratio” and “Ratio%” in table 1?
What is the unit of stand density in table 1? Number of trees per hectare or number of trees per square meter?
Line 145à What you mean by root drilling? You mean core sampling?
Line 147à How do you sample soil? Explain it
It is not clear that you did root sampling twice or once? You mentioned once in line 146 and kept those in the fridge and once by ingrown soil core? Please clarify
Line 163à use the same units in the manuscript. It is better to use all in cm or mm or m
Table 2à you explain the RTD, RLD, and ... once before. It is not necessary to do it again.
Line 250à “35302.42 m·m-3 and 36.81 m2·m-3” which unit is correct?
Line 305à “(Figure 4) and correlation analysis (Table 4)” you mentioned figure 4 at first, so you must show it before table 4.
Check all the units in the manuscript
Why SR7 behave completely different from others? What are the reasons? Discuss it.
What was your method for measuring richness? You measured only by number of species? Explain in methods.
Section 4.3à Are you discussed in this section? There are no references and comparing or discussing with other studies
You need to strengthen the discussion in all sections. Also, explain the novelty and implication of your study.
Author Response
Dear reviewer:
Thank you very much for reviewing our manuscript “Effects of Tree Species Diversity on Fine Root Morphological Characteristics, Productivity and Turnover rates ” (forests-1939872) and providing insightful comments and invaluable suggestions. These opinions are necessary and indispensable for the improvement of this research. Based on your comments and suggestions, we have revised our manuscript and prepared the point-by-point responses to your comments below. Finally, I sincerely hope that this article can be accepted by this journal, looking forward to your reply.
Comments and Suggestions for Authors
The manuscript entitled “Effects of Tree Species Diversity on Fine Root Morphological Characteristics, Productivity and Turnover rates” is assessing effect of tree species diversity on fine root productivity, turnover rate and morphological characteristics. The manuscript is interesting for the readers, however, it is not a complete paper, so it must improve and needs major revision. The most important part that must improve is discussion and novelty of the paper. Several major and minor comments that help to improve the paper as follow:
Line 40à what is he role of roots in ecosystems?
Response: the function of fine roots in ecosystems has been described in the revised manuscript. Showing as follows:
Fine roots (φ≤2 mm) also play important roles in carbon allocation and nutrient cycling in ecosystems.
Line 41-43à Several studies conducted about the roots and their different roles in ecosystem such as slope stability, hydrology, Biomass, etc. So, it is right that we still have a lack of knowledge, but it is not a good novelty for your study.
Response: The significance of fine roots on carbon allocation and nutrient cycling in forest ecosystems was highted in Line 40. The sentence shown in Line 41-43 illustrated the influence of roots on productivity and turnover rate of ecosystem. Both were used to elicit the main topic of this study. In addition, the role of roots in slope stability, hydrology, biomass, etc. was not illustrated, in order to keep major resrarch topic.
Line 44à reference?
Response: Corresponding reference has been added.
Line 54à Only soil nutrients? Influenced by biotic and abiotic.
Response: Both abiotic and biotic factors influence functional characteristics of fine roots, and these two factors are illustrated in Line 53-69 and Line 70-79, respectively.
Line 55-56à distribution of roots depends on different factors such as species age, vegetation species (trees, shrubs, herbaceous), topography, growth conditions (including soil temperature, soil depth, nutrient), and moisture content etc. Here you can add these parameters and cite the most recent studies such as:
https://doi.org/10.3390/f11030345
https://doi.org/10.1016/j.ecoleng.2017.04.034
https://doi.org/10.1007/s40333-018-0021-2
https://doi.org/10.3390/f10040341
https://doi.org/10.1016/j.foreco.2020.118873
https://doi.org/10.33494/nzjfs502020x68x
https://doi.org/10.1016/j.catena.2022.106410
Response:All references mentioned above have been added.
Line 94-95à Reference?
Response: Corresponding references have been added.
Line 99-102à Actually, root morphology also depends on the topography and genetic or tree species. It is not true that species richness is the only parameter that affect root morphology.
The novelty of the study is not clear. Clarify it.
The main and secondary aims of the study is not clear.
Response: the main purpose of this study was to investigate the effect of tree species diversity on fine root morphological characteristics, productivity and turnover of fine roots. Because the flat topography of experimental sites, the effect of topography on the morphological characteristics of fine roots was consistent in this study. Different tree species and genetic characteristics were certainly important factors affecting the morphological characteristics of fine roots, which were discussed in the second half of this manuscript.
In this sduty, the influence of tree species diversity on both morphological characteristics and turnover rate of fine roots was investigated in depth. In previous researches, the description about the influence of tree species diversity on productivity mainly focused on the above-ground parts of plants, while the study from the underground roots were limited.
Line 122-122à Did you do a soil experiment? If yes, explain the methods of sampling and number of samples. If not, how do you have the values of different soil properties?
Response:The data shown in Line 122-125 could be supported from references below, which have been added in revised manuscript.
- Jiang, H.; Du, H. Y. ; Bai, Y,Y.; Hu, Y.; Rao, Y.F.; Chen, C.; Cai, Y.L. Effects of spatiotemporal variation of soil salinity on fine root distribution in different plant configuration modes in new reclamation coastal saline field. Environmental Science and Pollution Research. 2016, 23, 6639–6650.
Jiang, H. Spatial and temporal distribution of fine root and its influencing factors research in plantation of coastal salt land. East China Normal University. 2016. (in Chinese)
Line 131à What are the species?
Response:Tree species and community characteristics were described in Table 1. Although there are great deal of tree species in our study, some of them are not our experimental objects, so the relative illustration was removed in the revised manuscript.
Line 136à “As shown in Table 1.” Do not use short sentences.Why write the species scientific name as a note again?
Response: Both short sentences and the note have been removed in the revised manuscript.
Line 139à How is it possible that herbs do nor grow in the soil?
Response: Owing to huge coverage, the lack of sunlight made it difficult for grass to grow in the community. In addition, the author also cleaned up the herbs in the community from time to time, so as to avoid the influence of these grass on roots of woody plants, which could affect the experimental results.
What are the “ratio” and “Ratio%” in table 1?
Response: The meaning of ratio and ratio% is same, so the ratio has been deleted in the revised version.
What is the unit of stand density in table 1? Number of trees per hectare or number of trees per square meter?
Response: Number of trees per hectare (Hong Jiang, 2016). The unit of stand density number of trees per hectare. This unit could be found from Hong Jiang group’s two articles: ‘Effects of spatiotemporal variation of soil salinity on fine root distribution in different plant configuration modes in new reclamation coastal saline field’ and ‘The spatial and seasonal variation characteristics of fine roots in different plant configuration modes in new reclamation saline soil of humid climate in China’, which were published in Environ Sci Pollut Res and Ecological Engineering, respectively.
Line 145à What you mean by root drilling? You mean core sampling?
Response: Root drilling means core sampling which was stated by Hong Jiang, et. al (2016).
The new description below could be found in the revised version.
In January, July and November 2018, fine roots were sampled with a steel bucket-type soil auger (φ=5 cm,H=30cm) with a T-handle in each plot. The sampling method is "S" shaped 9-point sampling, and the sampling depth is 50 cm.
Line 147à How do you sample soil? Explain it.It is not clear that you did root sampling twice or once? You mentioned once in line 146 and kept those in the fridge and once by ingrown soil core? Please clarify
Response: Root samples and soil were taken out once a month, packaged separately and then stored in a refrigerator. Other roots and soil samples collected by in-growth method were not be mixed with the above samples when stored in the fridge.
Line 163à use the same units in the manuscript. It is better to use all in cm or mm or m.Check all the units in the manuscript.
Response:Millimeter is the most commonly used unit for fine roots in academic area. However, for soil depth, the common unit is centimeter as shown in some published articles. The unit of fine root index adopts the unit commonly used in the articles published by other scholars.
Table 2à you explain the RTD, RLD, and ... once before. It is not necessary to do it again.
Response:Corresponding notes have been removed.
Line 250à “35302.42 m·m-3 and 36.81 m2·m-3” which unit is correct?
Response: m·m-3 is the unit of RLD, while m2·m-3 is the unit of RSAD.
Line 305à “(Figure 4) and correlation analysis (Table 4)” you mentioned figure 4 at first, so you must show it before table 4.
Response: The order of Figure 4 and Table 4 has been adjusted in the new version.
Why SR7 behave completely different from others? What are the reasons? Discuss it.
Response: SR7 has the most abundant tree species. Due to the increase of tree species, the competition between roots is intensified, and the root growth depth of different tree species is deeper, which promotes the change of the ecological niche of the root system. At the same time, because of the increase in tree species diversity, the biomass of fine roots increases and the productivity increases. In addition, the diversity of tree species in the community increased, and the competition between roots increased, which promoted the increase of dead fine roots and accelerated the turnover of fine roots to a certain extent.
What was your method for measuring richness? You measured only by number of species? Explain in methods.
Response: In this study, the richness is measured mainly according to the number of tree species and this test methods has been mentioned by Li, Y (2019) and Zeng W (2020) groups, respectively.
Section 4.3à Are you discussed in this section? There are no references and comparing or discussing with other studies.You need to strengthen the discussion in all sections. Also, explain the novelty and implication of your study.
Response: The relative references have been added in the revised manuscript to enhance comparisons with other scholars’ findings.
Reviewer 2 Report
Dear authors,
This manuscript adds to the little information that is available on fine root dynamics and is therefore of interest to the community. It is generally well written but requires some extra checks. Should the authors agree, the manuscript might be strengthened by a more thorough discussion.
Kind regards,

Author Response
Dear reviewer:
Thank you very much for reviewing our manuscript “Effects of Tree Species Diversity on Fine Root Morphological Characteristics, Productivity and Turnover rates ” (forests-1939872) and providing insightful comments and invaluable suggestions. These opinions are necessary and indispensable for the improvement of this research. Based on your comments and suggestions, we have revised our manuscript and prepared the point-by-point responses to your comments below. Finally, I sincerely hope that this article can be accepted by this journal, looking forward to your reply.
Comments and Suggestions for Authors
Dear authors,
This manuscript adds to the little information that is available on fine root dynamics and is therefore of interest to the community. It is generally well written but requires some extra checks. Should the authors agree, the manuscript might be strengthened by a more thorough discussion.
Kind regards,
Although the findings in this study might seem somewhat obvious, the findings do add to our limited knowledge concerning fine root dynamics. This article is generally well written and this study provides a clear conclusion that ‘species richness is positively correlated with fine root productivity’.
General suggestions
It seems, however, that this study might benefit from a more critical discussion on the overall study design, rather than a discussion of only the statistical results and their potential biological meaning. For example, given the difficulties in acquiring measurements of fine roots, how do you think the sample size, the measurements methods and statistical methods might have affected your outcomes? Does this change your interpretation of the results? Might spatial autocorrelation have affected your statistical results? Given the difficulties with measurements in the field, is p < 0.05 sufficient here, or would you cope for using a lower p value? How might this have affected your results?
Response: Multiple samplings were carried out over a year with sufficient replicates in each sampling. The scientific analysis and measurement methods of soil and root samples and accurate statistical analysis methods ensure the scientificity of the experimental results. The use of p < 0.05 for experimental precision and statistics is acceptable. The root systems of different tree species are quite different. At the same time, factors such as tree species diversity and soil physicochemical properties will affect the characteristics of the root system. Therefore, it is difficult to ensure p < 0.05 under the collection and treatment of a large number of root systems. Results showed that when p < 0.05, there was a significant difference in root characteristics, which was scientific. If a smaller value of p is used, the statistical results should be the same and will not affect the experimental results .
Although the text is generally well written there are some minor aspects that still needs some change. For example, the font size changes throughout the text and the spacing is inconsistent (e.g. L. 19, 67, 233, 239, 279, 280, 352,…, et cet.).
Response: All minor mistakes have been corrected in the manuscript.
Specific suggestions
Abstract
- L. 11 & 40 Is the simple, but common definition of fine roots being just less than 2 mm appropriate here? Might this have affected your results?
Response: Defining φ≤2 mm roots as fine roots does not affect the experimental results. In the previous articles, most of the root systems with φ≤2 mm were defined as fine roots. The definition was used in this paper to compare with the research results of other experts, so as to increase the researching consistency between different authors. Specific references have been listed in the revised version.
Introduction
- L. 30 It seems there is some word missing here.
Response: The minor mistake has been corrected in the manuscript.
- L. 46 Please remove ‘certain’ here or clarify a ‘certain impact’.
Response: The sentence ‘However, it is not clear what the impact will be’ has been deleted.
Methods
- L. 144 It seems that title 2.2. Sequential coring should probably be part of the next page.
Response: Corresponding change has been made in the new version.
- L. 145 Is there a reference for the “S” shaped 9-point sapling method?
Response: Corresponding references have been added in the revised version.
- L. 167 – 176 These sentences read like part of a laboratory manual. Could the authors rephrase them?
Response: The methods shown in Line 167-176 was mainly used in our laboratory, which provided a reference for future scholars to process and identify plant roots, and also ensured the accuracy of experimental results. This approach could be found from Hong Jiang’ group (2016).
Results
- Could you elaborate on why there was no effect of soil depth on RTD, SRL and SSA. I’ve noticed no in depth discussion on the lack of effects of the soil depth…
Response: This part mainly analyzed the influence of tree species diversity (biological factors) on morphological indicators, productivity and turnover rate of fine roots. The influence of abiotic factors characteristics on fine roots were mainly discussed in chapters 3.3 and 4.3.
Discussion
L. 340 Conformational plasticity Is mentioned here for the first time. Could you briefly clarify this word or alter the wording?
Response: Something has been changed in the revised version. Details as follow:
Plant does not respond to root competition through changing the morphological characteristics of fine root……
L. 342 Rather than referring to the first hypothesis, I believe it would aid the reader to just briefly mention the first hypothesis again.
Response: The first hypothesis has been re-described in the revised version.
L. 392 – 393 ‘The lifespan of fine roots has the strongest effect on its turnover’. Could you specify that you are referring to the carbon and nutrient turnover rather than ‘its’ turnover.
Response: The sentence ‘The lifespan of fine roots has the strongest effect on its turnover' has been deleted because the lifespan of fine roots was not explored, nor was the carbon and nutrient turnover characteristic of fine roots mentioned in our study.
Round 2
Reviewer 1 Report
Dear authors,
You did a good job! congratulation!